# Single-Cell RNA-Sequencing: Astrocyte and Microglial Heterogeneity in Health and Disease

**DOI:** 10.3390/cells11132021

**Published:** 2022-06-24

**Authors:** Michael S. Spurgat, Shao-Jun Tang

**Affiliations:** 1Department of Anesthesiology, Stony Brook University, 101 Nicolls Rd, Stony Brook, New York, NY 11794, USA; michael.spurgat@stonybrookmedicine.edu; 2Institute for Translational Sciences, University of Texas Medical Branch, 301 University Boulevard, Galveston, TX 77555, USA

**Keywords:** RNA-sequencing, astrocyte, microglia, HIV-1

## Abstract

Astrocytes and microglia are non-neuronal cells that maintain homeostasis within the central nervous system via their capacity to regulate neuronal transmission and prune synapses. Both astrocytes and microglia can undergo morphological and transcriptomic changes in response to infection with human immunodeficiency virus (HIV). While both astrocytes and microglia can be infected with HIV, HIV viral proteins in the local environment can interact with and activate these cells. Given that both astrocytes and microglia play critical roles in maintaining neuronal function, it will be critical to have an understanding of their heterogeneity and to identify genes and mechanisms that modulate their responses to HIV. Heterogeneity may include a depletion or increase in one or more astrocyte or microglial subtypes in different regions of the brain or spine as well as the gain or loss of a specific function. Single-cell RNA sequencing (scRNA-seq) has emerged as a powerful tool that can be used to characterise these changes within a given population. The use of this method facilitates the identification of subtypes and changes in cellular transcriptomes that develop in response to activation and various disease processes. In this review, we will examine recent studies that have used scRNA-seq to explore astrocyte and microglial heterogeneity in neurodegenerative diseases including Alzheimer’s disease and amyotrophic lateral sclerosis as well as in response to HIV infection. A careful review of these studies will expand our current understanding of cellular heterogeneity at homeostasis and in response to specific disease states.

## 1. Introduction

Until recently, researchers in neuroscience fields focused their efforts primarily on neurons and performed experiments designed to elucidate their structure, function, and responses to injury and infection. By contrast, the roles of non-neuronal cells and tissues have been largely overlooked. Although Rudolf Virchow discovered and provided the first description of glial cells as early as the mid-1800s, these cells were believed to function solely as support for essential neuronal functions. Thus, many of the unique properties and critical functions of glial cells remain to be explored. 

Various types of glial cells identified within the central nervous system (CNS) (i.e., microglia, astroglia [astrocytes], Bergmann glia, Müller glia, radial glia, and oligodendrocytes) exhibit different functions, localisation, and morphology. For example, microglia have been identified as immune cells that promote phagocytosis of foreign material and cellular debris and facilitate synaptic pruning. By contrast, astrocytes perform surveillance and regulate neurotransmission, whilst oligodendrocytes myelinate axons and radial glia serve as progenitor cells [1,2]. Given the critical homeostatic roles played by these cells, it is critical to understand modifications to glial structure and function that develop in response to disease.

Astrocytes and microglia undergo functional and transcriptomic changes in association with neurodegenerative disorders that include Alzheimer’s disease (AD), amyotrophic lateral sclerosis (ALS), and HIV-associated chronic pain [3,4,5,6]. Whilst different regions of the brain (e.g., the amygdala or nucleus accumbens) and spinal cord (e.g., the spinothalamic tract) contribute to the transmission and processing of neuronal signals, the contributions of astrocytes and microglia within these regions remain unclear. Furthermore, the aforementioned regions within the CNS include varied and heterogeneous populations of both astrocytes and microglia [7,8]. Thus, it will be important to determine the precise location, number, and unique function of each of these cell subtypes within the CNS as well as their contributions to region-specific damage associated with various disease states and the development of symptoms. Thus, a more comprehensive understanding of glial heterogeneity will be needed as a first step towards identifying populations with dysregulated gene expression as potential therapeutic targets.

The goal of this review is to discuss microglial and astrocyte heterogeneity at homeostasis and in response to disease conditions with a particular focus on results from recent studies that feature the use of single-cell (sc) or single-nucleus (sn) RNA sequencing (scRNA-seq).

## 2. Microglial Heterogeneity

Microglia are primary components of the immune regulatory system of the CNS. However, in contrast to macrophages and other immune cells, microglia do not originate in the bone marrow. Studies carried out in mouse models revealed that microglia originate from myeloid progenitor cells during early embryonic development [9]. Although macrophages may cross into the CNS, microglia are self-renewing within the CNS and do not ordinarily require replenishment from myeloid sources in the periphery. Microglia are distributed unevenly throughout the brain [8,10,11,12,13,14].

Microglial morphology and function are closely related phenomena. Microglia are highly ramified in the absence of external stimulation. Whilst the microglial body is comparatively small, several large protruding processes increase its surface area and facilitate sampling to detect changes in the environment. Microglia are highly motile when in this ramified state. In the absence of disease or injury, microglial processes undergo rapid and frequent retraction and growth [15].

### 2.1. General Biological Functions of Microglia

Microglia perform multiple roles within the CNS. While they are best known for their capacity to migrate through the CNS to facilitate the detection of foreign material and cellular debris, they are also capable of antigen presentation, phagocytosing myelin, and providing support for neural stem cell proliferation [15,16,17,18]. Microglia are guided towards injured or dead cells via their capacity to sense and respond to chemical gradients of molecules such as adenosine triphosphate (ATP) or lipopolysaccharide (LPS). This is typically followed by phagocytosis of synapses, foreign material, bacteria, and/or other agents identified at the sites of injury [19]. Each microglial activity profile relates directly to the specific cell subtype.

### 2.2. Microglial Subtypes: Molecular Signatures, Activity, and Function

The task of identifying specific microglial subtypes is somewhat difficult and complex largely because of their overlapping patterns of gene expression. This problem is further complicated by the fact that these profiles also overlap with those of macrophages infiltrating from the periphery. Microglia have been identified in immunofluorescence and fluorescence-activated cell sorting studies via their expression of specific proteins, including ionised calcium-binding adapter molecule 1 (IBA1), transmembrane protein 119 (TMEM119), and purinergic receptor P2Y12 (P2RY12) as well as cluster of differentiation (CD) markers CD16, CD68, and CD11b [20,21,22,23,24]. Similarly, gene markers including *Cx3cr1*, *P2ry12*, and *Tmem119* have been used to identify microglia in experimental studies featuring single-cell RNA sequencing (scRNA-seq) [22,23].

The results of a recent scRNA-seq study published by Zheng et al. [6] revealed distinct microglial subtypes within the brain and spine of wild-type mice at homeostasis with distinct transcriptomes that remained stable at two, four, and eight months of age. However, the authors reported only a partial overlap between the transcriptomes of microglia in the brain and spine. These results suggested the existence of unique populations of microglia in these locales. Similarly, whilst results from an earlier microarray study revealed significant differences in gene expression when comparing microglial populations in the cerebellum, cortex, hippocampus, and striatum [25], a subsequent study in which single-cell deep sequencing was to examine regional heterogeneity of mouse microglia reported no significant differences in these same regions [26]. Additional studies will be needed to resolve this discrepancy.

In addition to their region specific subtype heterogeneity, microglia can be activated via numerous proinflammatory pathways. These pathways can be initiated via microglial exposure to pathogen-associated molecular patterns (PAMPs) such as LPS and/or via damage-associated molecular patterns (DAMPs) that include free DNA fragments and extracellular ATP [27,28,29]. Activation of microglia by inflammatory mediators such as LPS initiates the release of ATP and proinflammatory cytokines including interleukin (IL)-1β and tumor necrosis factor (TNF)α to recruit additional immune cells to the affected area (reviewed in [30]). Once activated, microglia become less ramified and assume a more amoeboid shape. Amoeboid microglia have been associated with numerous conditions, including infections and AD [31,32].

The use of scRNA-seq approaches has expanded our understanding of microglia both at homeostasis and in various disease conditions. Microglia have been frequently classified using the well-established binary system in which microglia expressing inflammatory genes are considered M1, whilst M2 microglia are associated with a neuroregenerative phenotype. M1 microglia are defined as cells that express proinflammatory cytokines including TNFα, IL-1β, and IL-6, whilst M2 microglia are identified by the expression of TGFβ. Some investigators have divided these two classifications into additional subgroups based on gene expression patterns [24]. However, with the application of scRNA-seq, studies that classify microglial subtypes based on their functional differences are now feasible. With this method, microglia that promote increases in myelination based on lipid metabolism can be identified; this extends our understanding of these cells beyond the aforementioned proinflammatory or neuroregenerative classifications.

## 3. Microglial Subtypes Associated with the Pathogenesis of Specific Diseases

Studies featuring the use of sc/snRNA-seq have characterised transcriptomic changes in microglia over time and in response to various diseases including AD, ALS, and HIV infection (summarized in Table 1).

### 3.1. Alzheimer’s Disease (AD)

AD is a primary cause of dementia and is characterised by the accumulation of amyloid-beta in the brain. Based on current estimates, AD affects over 50 million people worldwide and is associated with an economic burden of one trillion dollars [37]. Although there is no cure for AD, there are currently several treatment options available that can be used to manage the symptoms, including cholinesterase inhibitors and antagonists of N-methyl-D-aspartate (NMDA) receptors [37]. Interestingly, recent studies have revealed that microglia aggregate near amyloid-beta plaques in the brain and potentiate disease severity [38,39].

Several publications describe the results of scRNA-seq studies designed to explore microglial heterogeneity in AD. Collectively, these studies have identified multiple transcriptionally-distinct microglial subtypes that are unique amongst patients with AD. Mathys et al. [33] used a scRNA-seq approach and identified the increased expression of *Ftl*, *Spp1*, *Slc11a1*, *Rpl28*, *Tmem163*, and *C1qc* over time in microglia from patients with AD-associated pathology and cognitive decline. These findings were associated with significant alterations in pathways related to neutrophil-mediated immunity, protein folding, the intrinsic apoptotic signaling pathway, and the regulation of protein stability in the distinct AD-related microglial subtypes; upregulation of these pathways is indicative of an increase in cellular stress responses. Similarly, Keren-Shaul [34] presented the results of scRNA-seq experiments in which microglia from wild-type mice, 5XFAD mice (a model for AD), and AD patients were evaluated. The results of this study revealed a specific subtype of microglia associated with neurodegeneration that was marked by increased expression of *Cst7* and *Lpl*, suggesting specific increases in both phagocytic activity and lipid metabolism. The results of this study also suggested a link between upregulated expression of these genes and a two-step process of *Trem2* activation in these microglia. Furthermore, these microglial subtypes may be associated with a larger array of neurodegenerative conditions as they were also found in association with Amyotrophic Lateral Sclerosis (ALS).

### 3.2. Amyotrophic Lateral Sclerosis (ALS)

ALS is a disease in which muscle atrophy ultimately leads to respiratory failure typically within several years after the initial diagnosis [40]. Similar to Alzheimer’s Disease (AD), several ALS-specific microglial subtypes have been identified, including cells that display unique ALS-specific morphology [41]. A study in which scRNA-seq was used to evaluate gene expression in all cells in the brainstem in the superoxide dismutase (SOD)1 overexpression mouse model of ALS revealed several known and previously-unidentified genes that were associated with this disorder [35]. Using this method, the authors highlighted the potential roles of inflammatory and synapse organisational pathways in microglia in this model; they also confirmed a role for peptidyl lysine modification in microglia and identified the regulation of Rac protein-mediated signal transduction. Whilst this study identified new pathways via which microglia may contribute to the pathogenesis of ALS, additional research will be needed to determine the significance of these findings. Overall, scRNA-seq is clearly a powerful tool that can be used to identify pathways that may be involved in neurodegenerative conditions.

### 3.3. Microglial Responses to HIV Infection

#### 3.3.1. Microglia as the Primary Target of HIV Infection in the CNS

HIV infects immune cells and ultimately leads to the destruction of the host’s immune system unless identified promptly and treated continuously. HIV virions initially target CD4-positive immune cells; these infected cells produce and release progeny virions, thus permitting the infection to spread. Infected cells can enter the CNS via transit through the blood-brain barrier, where they release nascent HIV particles that proceed to infect immune cells in the brain. Microglia express CD4 and other HIV co-receptors, including CCR5 and CXCR4, and thus can be infected with HIV [42]. Additionally, given that microglial cells may persist for many years, these cells represent a reservoir for HIV within the CNS [43]. Although there are now effective treatment options that can be used to manage HIV infection, people living with this infection frequently develop chronic pain and neurocognitive deficits [44,45]. Collectively, these findings strongly suggest that microglia may make critical contributions to persistent HIV-mediated pathology. Thus, studies focused on scRNA-seq of this cell population may provide insight into disease markers and pathways that may be targeted to delay the onset and/or limit the severity of these detrimental symptoms. 

#### 3.3.2. Microglial Subtypes Targeted by HIV Infection and/or Viral Proteins 

Two new microglial subtypes were identified in the brain and spinal cord of the gp120 transgenic mouse model of HIV-related pain by scRNA-seq [6]. A subtype that predominated in the gp120 transgenic mice, representing 11% of the total microglia evaluated, expressed elevated levels of *Lpl*, *Cst7*, and *Igf1* and displayed overall increased expression of genes associated with both demyelination and remyelination. This result suggests that this microglial subtype may promote altered myelination within the CNS and thus contribute to the development of pain. Another microglial subtype identified in the gp120 transgenic mouse model was classified as interferon-related and was characterised by increased expression of *Ifit2*, and *Ifit3*. These genes are associated with the activation of cellular interferon response and viral clearance [46,47]. Gene pathway analysis revealed that both the myelination-associated and interferon-related microglial subtypes in the spine and brain expressed significantly increased levels of genes associated with neuroinflammation [6], and thus another novel microglial mechanism that may contribute to the development of HIV-associated chronic pain. Another study featured the use of scRNA-seq and identified a rare subpopulation of myeloid cells in cerebrospinal fluid from an HIV-positive patient. Interestingly, the gene expression patterns detected in this rare myeloid subpopulation closely overlapped those identified in microglia identified in patients diagnosed with AD [36]. However, the transgenic gp120 model only expresses a single component of HIV, so further studies using human autopsies, HIV-infected humanized mouse models, or other animal models generated by infection with HIV-related virus such as ecoHIV, FIV (feline immunodeficiency virus) and SIV (simian immunodeficiency virus) will be useful for evaluating the findings [48,49,50,51]. Ongoing studies of the pathways in microglia infected and/or activated by HIV or its viral proteins may provide insight into new avenues for the treatment of chronic pain and neurocognitive deficits in these patients. 

### 3.4. Activated Microglia: Protective or Damaging?

Microglial activation serves an important immunological function in the CNS. However, prolonged activation can lead to neurodegeneration and disease progression. Once they have detected an injury or infection, microglia move to the affected area and proceed to phagocytose cellular debris and infectious particles. However, if the microglia are unable to resolve the injury, they may remain in a prolonged state of activation and continue to release proinflammatory cytokines. These responses can lead to a prolonged inflammatory state and may exacerbate existing symptoms. Therefore, whilst microglial activation is necessary for neuronal maintenance and defense against infection, prolonged activation may cause or exacerbate neurodegeneration.

## 4. Astrocyte Heterogeneity

Astrocytes are non-neuronal cells that are derived from radial glial cells in the embryonic ectoderm [52,53]. The morphology of a given astrocyte is based largely on its location within the CNS. Protoplasmic astrocytes found in the gray matter have several large branches protruding from the cell body with many finer branches emanating from each large branch. By contrast, fibrous astrocytes located in white matter exhibit many smaller processes that originate from the cell body. 

### 4.1. General Biological Functions of Astrocytes

Astrocytes perform numerous functions within the CNS. Historically, astrocytes were believed to contribute structural support and served as indicators of neurological disease. However, recent studies have shown that, in the absence of disease or injury, astrocytes may extend processes to cover specific regions of the brain and perform a variety of functions that regulate neuronal homeostasis. In studies in which electron microscopy was used to visualise their fine processes, protoplasmic astrocytes within the CA1 region of the hippocampus were found in non-overlapping domains. This finding suggests that astrocytes are capable of neuronal monitoring in distinct regions of the brain [54]. However, it is not yet clear whether this property is shared by fibrous astrocytes in white matter. Furthermore, results of other studies suggested that astrocytes may contribute to synaptic signaling; consistent with the tripartite synapse model, astrocytes may contribute to the glutamate/glutamine cycle together with neurons [55]. Astrocytes perform this function in many brain regions, including the cerebellum and cortex, via their capacity to sense and facilitate intake and removal of glutamate from the regions surrounding the synaptic cleft via the actions of GLT1, GLAST, and mGlur5 receptors located on their processes [56,57,58,59,60,61]. After uptake, glutamate is converted to glutamine within the astrocyte via the actions of glutamine synthetase and subsequently exported for reabsorption by neurons. Reabsorbed glutamine can be utilised directly or reconverted to glutamate for use as a neurotransmitter [62,63]. Astrocytes also facilitate neuronal calcium signaling. This signaling in astrocytes involves stimulation of their cell surface receptors to promote an increase of intracellular calcium via its release from the endoplasmic reticulum; alternatively, astrocyte activation may result in calcium influx via membrane receptors. The increase in intracellular calcium is then propagated simultaneously to nearby astrocytes through gap junctions [64]. Increases in intracellular calcium also lead to the release of ATP, gamma-aminobutyric acid (GABA), and glutamate [2]. Results from early studies designed to explore mechanisms leading to calcium signaling in astrocytes suggested that the release of these neurotransmitters may influence neuronal signaling [2,65,66]. However, the degree to which astrocytes participate in neuronal signaling remains uncertain. Also, because of the comparatively large distance between the synaptic cleft and astrocyte processes, it is conceivable that only high-affinity receptors with slow response times, for example, mGlur5, may be activated by this mechanism [67]. Because the response time of these targeted receptors may be too slow, astrocytes are unlikely to participate in neurotransmission. Astrocytes may also not be active participants in neurotransmission per se but may instead influence neurons via their capacity to influence synaptic plasticity [67].

### 4.2. Astrocyte Subtypes: Molecular Signatures and Activity

Astrocytes can be identified by their expression of specific marker proteins including glial fibrillary acidic protein (GFAP), S100 calcium-binding protein-β (S100β), aldehyde dehydrogenase family 1 member L1 (Aldh1l1), and excitatory amino acid transporter 2 (EAAT2/SLC1A2, or in mice GLT1) [68,69,70,71,72]. GFAP is an intermediate filament protein and is the prototypical marker used to identify astrocytes. GFAP localises within both the astrocyte cell body as well as in the larger processes. GFAP levels increase in response to injury or disease, leading to its use as a marker to identify specifically reactive astrocytes. However, it is critical to recognise that GFAP is expressed by other cell types, including radial glia progenitor cells [73,74]. The glutamate transporter, SLC1A2 is also found primarily in astrocytes where it plays a significant role in the removal of extrasynaptic glutamate [59,72]. The actions of this receptor protein contribute to the removal of 60–80% of the glutamate within the brain [59]. Mice devoid of SLC1A2 develop seizures and have a reduced lifespan, suggesting the importance of both this receptor and astrocytes in promoting neuronal maintenance [75]. Similarly, although markers such as GFAP can be used to identify the larger astrocytic processes, detection of the finer, smaller processes requires super-resolution or electron microscopy. It is critical to have some means of visualizing both the larger and the smaller processes to study the full interaction between astrocytes and neurons or blood vessels. 

Single-cell and single-nuclei studies of astrocytes (summarized in Table 2) have identified and characterised several subtype-specific functions across different regions of the brain; one study in wild-type mice identifies as many as five different subtypes of astrocytes [76]. Clustering analysis has confirmed previous findings and has confirmed regional astrocyte heterogeneity in comparisons between the cortex and the hippocampus [76]. Collectively, these findings suggest that there may also be functional differences in astrocytes within different regions of the CNS. Clustering analysis also identified three additional subtypes of mature astrocytes. This rigorous analysis revealed specific patterns of gene expression and distinct spatial localisation of each subtype within the brain. One subtype of mature astrocytes that was identified primarily within the hippocampus and subpial regions expressed comparatively higher levels of both *Gfap* and angiotensinogen (*Agt*). The increase in *Agt* expression suggests that this astrocyte subtype may contribute directly to synaptic plasticity and may interact closely with neurons [76]. The second subtype of mature astrocytes exhibited increased expression of *Unc13c* and was found primarily within the cortical layers. Although UNC13C is involved in the process of membrane fusion, additional studies will be needed to understand the specific role of this pathway in astrocytes. The third new subtype of mature astrocytes exhibited higher levels of *Agt* and reduced levels of *Gfap* and *Unc13c* expression and was found throughout the cortex and hippocampus. In conclusion, single-cell sequencing and clustering analysis can provide detailed insight into region-specific expression patterns that can be used to identify subtypes and, more importantly, presents a powerful tool for the identification and evaluation of specific changes that develop in response to disease.

## 5. Astrocytes in CNS Disease

Astrocytes undergo astrocytosis or astrogliosis in response to disease. This process involves morphological and transcriptomic changes in response to stimuli that include LPS, ischemia, and/or factors associated with diseases such as AD and ALS. Morphological signs of astrocytosis include hypertrophy and increased proliferation accompanied by the increased expression of *GFAP* [79]. For example, microglia stimulated by LPS release IL-1α, TNFα, and C1Q, all of which result in the conversion of astrocytes to an inflammatory state [80]. Cytokine release from activated microglia can promote this phenotypic change. This finding suggests that microglia and microglial-derived cytokines might be targeted for the development of therapeutics aimed at reducing or preventing the development of neuroinflammation. Similarly, neuronal ischemia or traumatic brain injury may also induce astrogliosis, which is a process that includes astrocyte hypertrophy and increased proliferation around the site of injury [79].

One scRNA-seq study that utilised >80,000 astrocytes compared the impact of LPS on these cells over time [77]. Amongst the findings, 68% of the astrocytes isolated from the LPS-treated mice expressed *SERPINA3N*, compared to only 3% in saline-treated mice. The results of this study suggest that SERPINA3N may be a marker for astrocyte activation in response to disease. Additionally, increased expression of genes associated with pathways that included “adaptive immune response” and “NF-κB signalling” were identified in astrocytes from LPS-treated mice compared to controls. This approach also revealed genes and pathways that were expressed at various time points or within specific regions of the brains of the LPS-treated and saline control mice. This in-depth analysis of the changes in gene expression and pathways again stands in support of the utility and power of the scRNA-seq approach.

### 5.1. Astrocyte Subtypes Associated with the Pathogenesis of Specific Diseases

#### 5.1.1. Alzheimer’s Disease (AD)

Results of a recent snRNA-seq study of human post-mortem tissue identified a cluster of astrocytes that were enriched in patients with AD characterised by increased expression of *GLUL* and *CLU*, the latter a previously-identified risk factor for developing AD [33]. Increased expression of genes including *MT2A*, *MT1E*, *GJA1* (CONNEXIN43), *LINGO1*, *RASGEF1B* were observed as were patterns associated with pathways involved in protein localisation to the endoplasmic reticulum, protein stabilisation, translational initiation, cellular response to mechanical stimulus, and neutrophil degranulation. Although further investigation will be needed to determine the significance of these findings, the amplification of pathways related to protein translation, localisation, and stabilisation suggest that this new astrocyte subtype may be more biologically active. The authors also reported a general decrease in the expression of genes associated with myelination; these findings suggest that astrocytes may be partially responsible for the demyelination observed during the progression of AD. Another snRNA-seq study targeting AD patients found that whilst the overall number of astrocytes remained unchanged, there was a significant decrease in the population of a neuroprotective astrocyte subtype in tissues from AD patients [78]. These results suggest that gene expression in astrocytes found in brain tissue from patients with AD may change transcriptionally to a non-neuroprotective phenotype; alternatively, astrocytes that die may be replaced with cells associated with the non-neuroprotective subtype. Also, astrocytes from the AD patient group exhibited decreased expression of *HES5*, *NTRK2*, *SLC1A2*, and *SPARCL1* in association with diminished synaptic signaling and glutamate secretion [78]. These latter results suggest that the depletion of astrocytes that contribute to the glutamate/glutamine cycle, a critical pathway that maintains neuronal homeostasis, may also contribute to the development of AD. Collectively, these studies demonstrate the utility of snRNA-seq for identifying changes in gene expression and pathways and the identification of potential therapeutic targets for the treatment of AD and related disease conditions.

#### 5.1.2. Amyotrophic Lateral Sclerosis (ALS)

Changes in astrocyte expression have also been reported in association with ALS. A scRNA-seq study targeting the brainstem in a mouse model for ALS revealed enrichment of previously identified pathways in astrocytes that are related to neurogenesis and CNS development. This study also identified novel pathways including one involved in the regulation and organisation of cell projections [35]. Results of this study revealed increased expression of *Mt1*, *Mt2*, and *Fam107a* and decreased expression of *Malat1* in astrocytes from this mouse model. While further studies will be needed to characterise the impact of these genes and their role in the pathogenesis of ALS, these results reveal the potential of scRNA-seq as a means to evaluate various disease states at the cellular and molecular levels.

#### 5.1.3. Astrocyte Reactions to HIV Infection

##### Astrocytes as a Target of HIV Infection in the CNS

Although astrocytes do not express the cellular receptors known to mediate direct infection with HIV, the results of numerous studies suggest that HIV can infect and ultimately activate these cells. Postmortem tissues from HIV-infected individuals revealed that 1–7% of astrocytes contained HIV DNA or mRNA [81,82]. These results suggest that very few astrocytes are HIV-infected and that most astrocytes do not support productive viral replication [83,84]. Additionally, astrocytes can remain HIV-infected for >60 days and can spread from astrocytes to other cells in the CNS or into peripheral circulation through gap junctions or mechanisms unknown [82,85,86]. As astrocytes can be HIV-infected and can spread the virus, we would certainly benefit from additional information on their role in disease progression and symptomatology.

##### Astrocyte Subtypes Associated with HIV Infection

One earlier study used microarrays to examine transcriptomic changes associated with immune and neuronal function in cultured human astrocytes that were HIV-infected or exposed to HIV protein gp120. Both conditions resulted in similar patterns of differential gene regulation, although HIV infection alone resulted in significantly increased expression of chemokine and cytokine-related genes [87]. Amongst the interpretations suggested, the differences in chemokine and cytokine gene expression may relate to contributions from other HIV proteins, including Tat and Nef. Another study used bulk RNA-sequencing to examine the impact of HIV virions on primary cultured human astrocytes. Exposure to HIV resulted in the enrichment of genes related to inducible nitric oxide synthase as well as signalling associated with TNF receptor 2 signalling, neuroinflammation, and interferon responses [83]. The increases in gene expression levels associated with these pathways indicate that HIV induces proinflammatory pathways in astrocytes in vitro and suggests that these cells may contribute to the development of neurocognitive impairment or chronic pain. Additional comparisons between HIV-treated astrocytes and astrocytes from patients with neurocognitive impairment revealed shared expression patterns featuring genes related to neuroinflammation and interferon signalling. These results suggest that the in vitro cell culture model recapitulates the transcriptomic changes exhibited by astrocytes from HIV patients and that these astrocytes may contribute to HIV-associated neurocognitive impairment. The study also identifies several potential mechanisms underlying astrocyte-mediated neurocognitive dysfunction. Collectively, these studies provide valuable insight into HIV-mediated changes exhibited by the astrocyte transcriptome that may ultimately lead to neurocognitive impairment and chronic pain. Future studies focused on results at the single-cell level may provide additional insight into HIV-mediated changes to astrocytes and have the potential to lead to the development of targeted therapeutics.

## 6. Concluding Remarks

Sc/snRNA-seq is a powerful approach for studying transcriptomic alterations in response to disease at the single cell resolution. This approach has the potential to identify hundreds of genes within a specific type of cell that are differentially regulated in response to specific conditions, including diseases (Figure 1). Compared with more ‘traditional’ platforms such as bulk RNA-seq and microarrays, sc/snRNA-seq has the unprecedented power to identify small groups of cells that are specifically associated with given physiological or pathological conditions. However, sc/snRNA-seq is currently associated with several shortcomings. It can only reveal the expression profiles of a small portion of a transcriptome, leaving many genes expressed at lower levels unidentified. Increasing the transcriptomic complexity revealed by sc/snRNA-seq has the potential to enhance the power to identify rare cell types. Another shortcoming of the current sc/snRNA-seq platforms is the lack of functional integration of the cells analyzed. It would be a marked advance if the cell types defined by the transcriptomes revealed by sc/snRNA-seq can be matched with specific physiological functions. 

## Figures and Tables

**Figure 1 cells-11-02021-f001:**
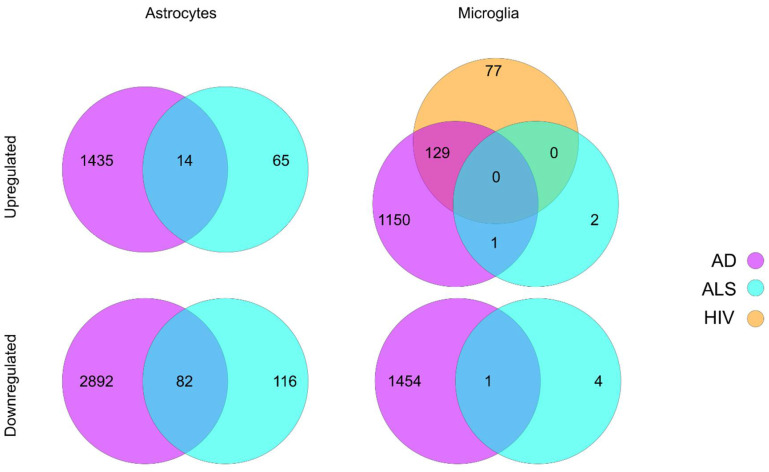
Summary of differential gene expression in astrocytes and microglia reported in sc/snRNA-seq studies of ALS, AD, and HIV. Lists of differentially expressed genes were acquired from the respective papers and filtered based on the reported adjusted *p* value [6,33,34,78]. Upregulated genes were identified based on a log fold change or z-score greater than zero, while downregulated genes were identified based on a log fold change of z-score less than zero.

**Table 1 cells-11-02021-t001:** Summary of findings identified from microglial RNA sequencing studies.

Experiment	Species & Location	Disease State	Marker Gene(S)	Pathways	Reference
Single-cell RNA-seq	Mouse spine, and brain cortex	HIV	*Ifit2, Ifit3*	Interferon Signaling	[6]
Single-cell RNA-seq	Mouse spine, and brain cortex	HIV	*Lpl, Cst7, Igf1*	Myelination	[6]
Single-nucleus RNA-seq	Human prefrontal cortex	AD	*FTL, SPP1, SLC11A1, RPL28, TMEM163, C1QC*	Neutrophil- mediated immunity, protein folding, intrinsic apoptotic signaling pathway, and regulation of protein stability	[33]
Single-cell RNA-seq	Mouse brain	AD	*Lpl, Cst7*	Phagocytosis, lipid metabolism	[34]
Single-cell RNA-seq	Mouse brainstem	ALS	*Camk2b*	Regulation of Rac protein signal transduction, peptidyl lysine modification	[35]
Single-cell RNA-seq	Human CSF	HIV	*C1QC, APOE, MSR1*	N/A	[36]

**Table 2 cells-11-02021-t002:** Summary of findings from astrocytic RNA sequencing studies. Marker genes in bold are increased. Marker genes that are only italicized are decreased. WT: Wild-type.

Experiment	Species & Tissue Location	Disease State	Marker Gene(S)	Pathways	Reference
Single-nucleus RNA-seq	Human prefrontal cortex	AD	* **MT2A, MT1E, GJA1, LINGO1, RASGEF1B** *	Protein localisation, protein translation, protein stabilisation	[33]
Single-cell RNA-seq	Mouse brainstem	ALS	***Mt1, Mt2, Fam107a**, Malat1*	Neurogenesis, regulation of cell projection organization, neuron differentiation	[35]
Single-cell RNA-seq	Mouse cortical layers	WT	* **Unc13c** *	Insulin signaling	[76]
Single-cell RNA-seq	Mouse cortex and hippocampus	WT	***Agt**, Gfap, Unc13c*	Transcription regulators	[76]
Single-cell RNA-seq	Mouse hippocampus	WT	* **Frzb, Ascl1** *	Mitosis and cell cycle regulation, neural tissue development	[76]
Single-cell RNA-seq	Mouse cortex and hippocampus	WT	* **Fam107a** *	Glucose metabolism, energy production, neural tissue development	[76]
Single-cell RNA-seq	Mouse hippocampus	WT	* **Gfap, Agt** *	Synaptogenesis, synaptic plasticity, glutamatergic neurotransmission	[76]
Single-cell RNA-seq	Mouse brain	LPS treatment	* **Serpina3n** *	Adaptive immune response, NF-kB signalling	[77]
Single-nucleus RNA-seq	Human prefrontal cortex	AD	* **ADGRV1, GPC5, RYR3** *	Synaptic signaling, glutamate secretion	[78]

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
