# Peer review of "Single-Cell RNA-Sequencing: Astrocyte and Microglial Heterogeneity in Health and Disease"

_cells, 2022, doi:10.3390/cells11132021_

Round 1
Reviewer 1 Report
Manuscript ID: Cells-1728482
Title: Single-cell RNA-sequencing: Astrocyte and Microglial Heterogeneity in Health and Disease
Cells
The manuscript reviews recent scRNA-seq studies that have explored microglia and astrocyte heterogeneity in Alzheimer’s disease, Amyotrophic lateral sclerosis (ALS), and HIV infection. Prior reviewing microglia and astrocyte heterogeneity in these three neurodegenerative diseases basic information about microglia and astrocytes’ biological function and subtypes are presented. The review has a defined outline, and the overall structure is logical. Each section concludes with reiterating the importance of scRNA-seq as a powerful tool to identify disease specific changes and new potential therapeutic targets for treatment intervention. To increase impact of the review it is suggested to add a future direction section at the end of the review and outline steps that need to be taken to move the research field forward (potentially also adding limitations of scRNA-seq that still need to be overcome). A summary graphic also would help the reader to understand the unique and overlapping findings generated by scRNA-seq for the three outlined diseases. Overall, this article provides a review of an important and timely topic and is very well written. However, its impact would be enhanced if more ideas on future directions/limitations of scRNA-seq are presented as well as a graphic is included that summarizes the overall findings. Specific suggestions are as follows:
Major Comments:
- Adding a future direction section at the end of the review to outline steps that need to be taken to move the research field forward (potentially also adding limitations of scRNA-seq that still needs to be overcome) would increase impact of the review. For example, in various sections the authors make the comment that the significance of scRNA-seq findings need to be determined by additional research. Even though in various sections the authors are trying to add a concluding sentence with a potential future direction it is more superficial; by adding a future direction section that outlines future directions more in depth the reader will be able to take away it’s importance, significance, and necessity (what is needed to continue to move the field forward)
- Subsection 3.2: More details on scRNA-seq findings need to be provided, such as specific genes that have been found to be altered in the referred studies; because information is much lacking here the concluding sentence of scRNA-seq being a powerful tool seems to be out of place
- Adding a summary graphic on the unique and overlapping findings generated by scRNA-seq for the three outlined diseases would help the reader to understand its contribution to neurodegenerative diseases in general and its unique contribution to each of the three diseases
Minor Comments:
- Pages 5-6; Title of subsections 4.1 and 4.2 should be italicized and not bolded (see Section 2)
- Page 3, line 114: “pro-inflammatory” – should read “proinflammatory”
- Page 3, lines 126 & 134: “neuro regenerative” – should read “neuroregenerative”

Author Response
We appreciate the reviewer’s comments regarding the impact and insight of the review paper. We also appreciate the reviewer’s comments and suggestions. After incorporating the feedback from reviewer 1,
we believe the revised review paper has improved and hope it is ready for publication.
Major Comments:
● Adding a future direction section at the end of the review to outline steps that need to be taken to move the research field forward (potentially also adding limitations of scRNA-seq that still needs to be overcome) would increase impact of the review. For example, in various sections the authors make the comment that the significance of scRNA-seq findings need to be determined by additional research. Even though in various sections the authors are trying to add a concluding sentence with a potential future direction it is more superficial; by adding a future direction section that outlines future directions more in depth the reader will be able to take away it’s importance, significance, and necessity (what is needed to continue to move the field forward)
Response: We agree with the point that a future direction section would increase the impact of the review. A conclusion (page 11, section 6) has been added at the end of the review. This section discusses pros, cons, and limitations of scRNA-seq.
● Subsection 3.2: More details on scRNA-seq findings need to be provided, such as specific genes that have been found to be altered in the referred studies; because information is much lacking here the concluding sentence of scRNA-seq being a powerful tool seems to be out of place
Response: We agree that the concluding sentence was inadequately supported by earlier statements in the paragraph. We have added genes which were found to be altered in the studies cited in this section (page 5, section 3.2).
● Adding a summary graphic on the unique and overlapping findings generated by scRNA-seq for the three outlined diseases would help the reader to understand its contribution to neurodegenerative diseases in general and its unique contribution to each of the three diseases
Response: A summary graphic (page 10, figure 1) has been added to the review. This graphic summarizes overlapping genes between the studies on ALS, AD, and HIV which are cited throughout the paper.
Minor Comments:
- Pages 5-6; Title of subsections 4.1 and 4.2 should be italicized and not bolded (see Section 2)
- Page 3, line 114: “pro-inflammatory” – should read “proinflammatory”
- Page 3, lines 126 & 134: “neuro regenerative” – should read “neuroregenerative”
All minor comments have been corrected.
Reviewer 2 Report
Spurgat and Tang have summarized recent studies that use single cell RNAseq (scRNA) to clarity the role of microglia and astrocytes in human diseases including neurodegenerative disorders and HIV infection. scRNAseq is a very powerful tool to elucidate transcriptional changes at the single cell level. Since scRNAseq does not require a large quantity of samples, it has been successfully used to study heterogeneity of CNS-resident cells from precious human in vivo samples such cerebrospinal fluid (CSF). The authors are focused on Alzheimer’s disease (AD) and amyotrophic lateral sclerosis (ASL) as well as HIV infection in this manuscript. This is a very comprehensive and interesting review and would provide insights into the role of microglia and astrocytes in these diseases. Here are some suggestions and minor points to be addressed:
1. It would be nice to have one paragraph (like Concluding Remarks) at the end to summarize pros and cons of scRNAseq studies. scRNA cannot detect epitranscriptomic RNA modifications, which can impact microglia and astrocyte biology. Also, prospects of scRNAseq analysis in CNS research would be appreciated. As for HIV research, studies using more relevant animal models for HIV infection (non-human primates, humanized mouse etc) and human primary samples are warranted.
2. The paragraph 3.2.2 discusses a study using gp120 transgenic mice. Although these findings are interesting, they need to be carefully interpreted since this system is fairly artificial. This limitation ought to be noted and use of more relevant animal models could be discussed.
3. Table 2 includes bulk RNAseq and microarray studies, which are not the main theme of this review.
Minor points:
- Table 1, first two studies: marker genes and pathways are not matched.
- Line 216, CCL12 seems not to be described in the original paper.
Author Response
We appreciate the reviewer’s comments regarding the impact and insight of the review paper. We also appreciate the reviewer’s comments and suggestions. After incorporating the feedback from reviewer 2, we believe the revised review paper has improved and hope it is ready for publication.
- It would be nice to have one paragraph (like Concluding Remarks) at the end to summarize pros and cons of scRNAseq studies. scRNA cannot detect epitranscriptomic RNA modifications, which can impact microglia and astrocyte biology. Also, prospects of scRNAseq analysis in CNS research would be appreciated. As for HIV research, studies using more relevant animal models for HIV infection (non-human primates, humanized mouse etc) and human primary samples are warranted.
Response: We agree that a conclusion section is warranted. A concluding remarks section (page 11, section 6) has been added. In this section, we review the pros and cons of scRNAseq and briefly discuss future prospects of sc/snRNA-seq.
- The paragraph 3.2.2 discusses a study using gp120 transgenic mice. Although these findings are interesting, they need to be carefully interpreted since this system is fairly artificial. This limitation ought to be noted and use of more relevant animal models could be discussed.
Response: We agree that this information should be included. This limitation of gp120 transgenic mice was briefly discussed and suggestions on alternative animal models are included in paragraph 3.2.2 (page 5).
- Table 2 includes bulk RNAseq and microarray studies, which are not the main theme of this review.
Response: These studies have been removed from the table (page 8, table 2).
Minor points:
- Table 1, first two studies: marker genes and pathways are not matched.
- Line 216, CCL12 seems not to be described in the original paper.
Response: All minor comments have been corrected.
Round 2
Reviewer 1 Report
With the revisions the manuscript has significantly improved. Thank you for the authors' responses to my comments.
Author Response
Point 1: With the revisions the manuscript has significantly improved. Thank you for the authors' responses to my comments.
Response 1: We thank the reviewer for their comment. We also thank them for their feedback which contributed to this improvement.